# Improving Tirapazamine (TPZ) to Target and Eradicate Hypoxia Tumors by Gold Nanoparticle Carriers

**DOI:** 10.3390/pharmaceutics14040847

**Published:** 2022-04-12

**Authors:** Giimel Ajnai, Chun-Chia Cheng, Tzu-Chun Kan, Jeng-Wei Lu, Sri Rahayu, Amy Chiu, Jungshan Chang

**Affiliations:** 1Graduate Institute of Medical Sciences, College of Medicine, Taipei Medical University, Taipei 11031, Taiwan; giimel.a@4hospital.mn (G.A.); d119102016@tmu.edu.tw (T.-C.K.); 2Department of Immunology, Mongolian National University of Medical Sciences, Ulaanbaatar 14210, Mongolia; 3Training, Research and Innovation Department, Fourth Hospital, Ulaanbaatar 13370, Mongolia; 4Radiation Biology Research Center, Institute for Radiological Research, Chang Gung University/Chang Gung Memorial Hospital, Taoyuan 33302, Taiwan; cccheng.biocompare@gmail.com; 5Antimicrobial Resistance Interdisciplinary Research Group, Singapore-MIT-Alliance for Research and Technology, Singapore 138602, Singapore; jengweilu@gmail.com; 6Department of Biology, Faculty of Mathematics and Natural Science, Universitas Negeri Jakarta, Jakarta 13220, Indonesia; srirahayu@unj.ac.id; 7Department of Translational Research and Cellular Therapeutics, Arthur Riggs Diabetes and Metabolic Research Institute, Beckman Research Institute, City of Hope, Duarte, CA 91010, USA; amchiu@coh.org; 8International Master/Ph.D. Program in Medicine, College of Medicine, Taipei Medical University, Taipei 11031, Taiwan; 9International Ph.D. Program for Cell Therapy and Regeneration Medicine, College of Medicine, Taipei Medical University, Taipei 11031, Taiwan

**Keywords:** hypoxia, Tirapazamine (TPZ), MKN45 cells, xenograft, bovine serum albumin, gold nanoparticles (GNPs)

## Abstract

Tumor hypoxia is a hallmark of solid tumors and emerged as the therapeutic target for cancer treatments, such as a prodrug Tirapazamine (TPZ) activated in hypoxia. To increase tumor accumulation, gold nanoparticles (GNPs) were selected to conjugate with TPZ. In this study, we successfully formulated and assessed the biochemical and therapeutic roles of the conjugated gold nanoparticles–Tirapazamine (GNPs–TPZ) on therapeutic assessments of MKN45-induced xenograft animal model. The results indicated that GNPs–TPZ was a potential nanomedicine for selectively targeting hypoxia tumors coupled with decreased side effects on healthy tissue or organs. TPZ significantly reduced cell viability of hypoxic gastric cancer MKN45 cells, but not in cells incubated in normoxia condition. For improving tumor targeting efficiency, furthermore, the GNPs drug carrier was conjugated to TPZ via biding mediator bovine serum albumin (BSA), and we demonstrated that this conjugated GNPs–TPZ retained the unique characteristics of hypoxic toxin and possessed the adequate feature of systemic bio-distributions in animals. GNPs–TPZ nanoparticles revealed their superior affinity to hypoxia tumors in the MKN45 xenograft. Moreover, GNPs–TPZ treatments did not significantly alter the biochemical parameters of blood samples acquired from animals. Taken together, TPZ, a prodrug activated by hypoxia, was conjugated with GNPs, whereas BSA severed as an excellent binding agent for preparing the conjugated GNPs–TPZ nanomedicines. We demonstrated that GNPs–TPZ enhanced tumor targeting, resulting in higher therapeutic efficacy compared to TPZ. We suggest that it may sever as an adjuvant treatment or combined therapy with other chemotherapeutics for the treatment of cancer patients in the future.

## 1. Introduction

Cancer is the one of major diseases causing the death of patients worldwide and is responsible for one in four deaths in the United States [1]. Hypoxia is a typical feature in most solid tumors and is strongly associated with tumor progression, immunosuppression, and drug resistance to therapy [2,3,4,5,6,7,8]. When tumors outgrow their environment, nutrient diffusion becomes insufficient, and a considerably low supply of required oxygen available to the inner region of tumors leads to the initiation of tumor hypoxia. Tumor cells in these hypoxic regions tend to become resistant to radiation therapy and anticancer drugs. The hypoxia-mediated resistance to chemotherapy possibly results from changed cancer cell metabolism, coupled with a slower proliferation rate, contributing to therapy resistance by inducing cell quiescence [9]. For these reasons, tumor hypoxia is considered one of the greatest challenges in treating solid tumors and also emerges as one of the best areas to target in cancer therapy [10,11].

Several hypoxia-activated prodrugs (HPA), such as Tirapazamine (TPZ), AQ4N, PR-104, EO9 (Apaziquone), TH-302 (Evofosfamide), and SN30000, were generated, and these bioreductive drugs selectively turn to be cytotoxic to hypoxic cells [12]. TPZ (3-amino-1,2,4-benzotriazine 1,4-dioxide) is a hypoxic cytotoxin that specifically damages oxygen insufficient tumor cells. It has been demonstrated that TPZ gains greater cytotoxicity to hypoxic cells than to oxygenated cells in vitro [13,14]. This aromatic compound was first prepared in a screening program for new herbicides. In 1986, its potential clinical use was first described by Zeman and colleagues, who illustrated the promising effect of toxicity towards hypoxic mammalian cells [15]. Under hypoxic conditions, this agent is bio-reduced to a nitroxide-based free radical that can attract hydrogen from DNA strands, causing a break in DNA [16].

To improve the therapeutic effects of TPZ, combined therapy using TPZ plus cisplatin (CCDP) has been extensively studied in clinical trials. The results revealed that combined therapy showed a promising therapeutic effect during clinical phase I and II, but failed to show improvement in the overall survival rate of patients with advanced-stage head and neck cancer or localized cervical cancer in the I clinical trial phase II [14,17,18]. In addition, this combined therapy led to unwanted severe side effects, such as gastrointestinal grade 3 or 4 toxicity, which mostly involves vomiting, nausea, diarrhea, and myalgia in patients [19,20,21,22]. Therefore, improvements in tumor-targeted delivery of TPZ is one of the most crucial tasks to avoid severe systemic side effects coupled with increased better prognostic outcomes in patients.

Gold nanoparticles (GNPs) were usually used as a vehicle for drug or gene delivery to the targeted tumors due to the unique feature of enhanced permeability and retention effect (EPR) [12,23,24]. In addition, as therapeutics vectors for drug or gene delivery, GNPs have also been applied in many potential biological applications, such as thermal actuators for cancer therapy, and agents for diagnostic imaging [25,26]. For nanoparticle conjugation, the mediator or binding agents are required for bridging GNPs to drugs or ligands. Previous reports have indicated that plasma proteins, such as bovine serum albumin (BSA), can be a great candidate to facilitate drugs, for example, the decoration of TPZ onto GNPs [27,28]. Albumin is a natural carrier for hydrophobic molecules to transport and distribute many biological molecules and drugs through the circulatory into targeted tissue or organs [29,30]. Currently, bovine serum albumin is widely used in drug delivery systems [31]. For example, BSA nanoparticles have been used in multilayer thin films for localized delivery of Doxorubicin (DOX), and showed a pH-dependent DOX release behavior [32]. Moreover, DOX was successfully loaded onto BSA nanoparticles and displayed a greater therapeutic effect in reductions of cancer cell viability than the free form of drugs [33,34].

In this study, we successfully manufactured conjugated GNPs–TPZ nanoparticles with the assistance of binding agent BSA followed by functional therapeutic assessment in human gastric cancer cells MKN45 and also in a MKN45-induced xenograft animal model. GNPs–TPZ displayed good tumor-targeted affinity coupled with enhanced therapeutic effects on MKN45 xenograft animals compared to TPZ alone. It demonstrated that TPZ remained maintains its therapeutic characteristics as a hypoxic toxin even in the form of GNPs–TPZ by triggering the apoptotic cell death program in hypoxia tumors. No obvious altered biochemical and inflammatory parameters were detected in animals, indicating the safety of GNPs–TPZ nanomedicines. Taken together, BSA-decorated GNPs were considered potential drug carriers of TPZ and the conjugated GNPs–TPZ displayed the specificity to target and eradicate hypoxic tumor cells.

## 2. Materials and Methods

### 2.1. Tumor Xenograft Mice

Male nude mice were obtained from National Laboratory Animal Center, Taiwan. Mice were selected and housed in a 12-h light cycle at 22 °C and given food (mouse standard diet) or water ad libitum. Animal experimentation was performed according to the approved procedures in the Institute of Nuclear Energy Research, Atomic Energy Council, Taoyuan, Taiwan. Tumor xenografts were established by injecting 2 × 10^6^ human gastric cancer cells, MKN45, subcutaneously into the legs of nude mice that were 6~8 weeks old. The tumor imaging experiment was performed 2 weeks after tumor cell injection.

### 2.2. Synthesis of Gold Nanoparticles

First, aqua regia was prepared by mixing three parts hydrochloric acid (HCl) to 1-part nitric acid (HNO_3_) by volume in a beaker. It was used to clean the glassware by dissolving any residual metallic particles which could possibly interfere with the synthesis of gold nanoparticles. Briefly, 1 mL of 0.1 M HAuCl_4_ was added into 500 mL ultrapure water and stirred vigorously. Then, 0.06 g of NaBH_4_ in 10 mL ultrapure water was added, using a pipette, and 10 mL of NaBH_4_ solution was transferred dropwise to the flask. The solution in the flask changed from yellow to ruby red. The ruby red color indicated the formation of gold nanoparticles (Appendix A). There was no obvious precipitation in the GNPs solution with or without BSA incubation. Absorbance exchanged from 520 to 534 nm after conjugation with BSA (Appendix A). 

### 2.3. Conjugations of Gold Nanoparticles with TPZ and Cy7.5

Albumin was used as a stabilizer for GNPs, and as a crosslinker to conjugate GNPs with other chemicals. For example, the conjugation of gold nanoparticles with TPZ was achieved by adding glutaraldehyde as a crosslinker to conjugate the individual amine groups (-NH_2_) from BSA and TPZ. The process of glutaraldehyde crosslinking amine groups was adapted and modified from a previous study [35]. In brief, 1 mL of 3 mg/mL BSA was pre-incubated with 1 mg of TPZ in 0.1 M carbonate–bicarbonate solution of pH 9.6. Then 10 μL of 25% glutaraldehyde (finally 0.25%) was added to the solution and allowed to react for 15 min at room temperature. The Sephadex G-25 column was used to purify the BSA–TPZ, which was eluted in 2 fractions by distilled water. Consequently, the BSA–TPZ was dried by using a freeze dryer (FreeZone Freeze Dry Systems, LABOCONCO, Kansas City, MO, USA) and then dissolved in 1 mL of synthesized gold nanoparticles for >2 h, at room temperature, with gentle shaking. Then the Sephadex G-25 column was used to purify the gold-nanoparticle-linked TPZ, using a PBS buffer. The concentration of obtained TPZ was 25 μg/mL, measured at OD 450 nm. In order to conjugate gold nanoparticles with Cy7.5, 0.5 mg of Cy7.5 and 3 mg/mL BSA were mixed in 0.1 M of carbonated-bicarbonate solution (pH 8.0), at room temperature, for 2 h, with gentle shaking to obtain the BSA–Cy7.5 product. The subsequent conjugation with GNPs used the same procedure as that of the GNPs–TPZ described above.

### 2.4. Characteristics of Synthesized Gold Nanoparticles

In addition to the observation of colorful change directly in GNPs solution, the successful conjugation of GNPs with TPZ was measured by using a UV–Vis Spectrophotometer (V-650, Jasco, Portland, OR, USA) to observe the increased wavelength. Moreover, the size of the GNPs was determined by using dynamic light scattering (DelsaNano S, Beckman Coulter, Brea, CA, USA).

### 2.5. Detection of BSA Conjugates Using MALDI–TOF MS

Albumin with or without TPZ was analyzed by using MALDI–TOF MS (UltraFlexIII, Bruker Corporation, Billerica, MA, USA). The instrument was in a positive linear mode at ~30% laser power with a standard 337-nmN2 laser. The protein samples were premixed with 20 mg/mL of matrix sinapic acid at a 1:1 ratio and then spotted on the steel plate to dry completely. The molecular weight of albumin was measured as ~66 kDa, which was shifted to 74.2 kDa after TPZ conjugation. The increased molecular weight of BSA indicated the successful conjugation with the specific chemical, such as TPZ, in this study.

### 2.6. Tumor Inhibition Assay In Vitro and In Vivo

MKN45 cells were cultured in 10 mL of DMEM medium with 10% FBS and 1% antibiotics (mixture of 100 U/mL of penicillin and 100 μg/mL of streptomycin) in a 5% CO_2_, 37 °C incubator for 48 h. The cell viabilities of MKN45 cells treated with GNPs or GNPs–TPZ were measured and confirmed by using tetrazolium salt WST-1 assay (2-(4-iodophenyl)-3-(4-nitrophenyl)-5-2, 4-disulfophenyl)-2H-tetrazolium (Takara, Mountain View, CA, USA). For tumor therapy, TPZ or GNPs–TPZ at 0.5 dose of mg/kg was injected 10, 14, and 17 days after transplanted tumors reached 50 mm^3^ (*n* = 3 for each group). The tumor volume was calculated as the following formula: length × width^2^ × 0.52.

### 2.7. TUNEL Assay

Surgical removal of liver and tumors from xenografts were kept in formalin-fixed and paraffin-embedded 4 μm sections for subsequent TUNEL assay, using a TUNEL assay system (Promega, Madison, WI, USA). The experiment was performed according to the manufacturer’s instructions.

### 2.8. In Vivo Imaging in Gastric Cancer Xenografts

An in vivo imaging system (IVIS, PerkinElmer, Waltham, MA, USA) was used to detect the targeted accumulation of GNPs. First, GNPs were conjugated with Cy7.5 fluorescence. After GNPs–Cy7.5 was injected via the tail vein for 48 h, the fluorescent intensity in tumor tissues was acquired and compared to that in muscle as a control baseline. The fluorescent expressions in organs captured from the individual tumor xenografts were detected by using the IVIS system (*n* = 3). In addition, the HypoxiSense 680 agent (PerkinElmer, Waltham, MA, USA) targeting carbonic anhydrase 9 was performed to detect hypoxia conditions in tumor xenografts. The images were acquired 24 h after injection of the HypoxiSense 680 agent.

### 2.9. Immunoblotting

Cells and tissues were lysed in the buffer containing 150 mM NaCl, 1% NP-40, 0.1% SDS, and 50 mM Tris-HCl (pH8.0). The 15 μg of protein samples was diluted in 2× Laemmli sample buffer (final concentrations: 75 mM Tris pH 6.8, 10% (*v*/*v*) glycerol, 2% SDS (*w*/*v*), and 0.002% (*w*/*v*) bromophenol blue), separated by the 10% sodium dodecyl sulfate polyacrylamide gel electrophoresis, and then transferred onto the Immobilon P membranes (Merck Millipore, Burlington, MA, USA). These membranes were blocked in the blocking buffer (Goal Bio, Taipei, Taiwan) for 1 min, at room temperature. Membranes were incubated with primary antibodies (diluted in 1 mL of blocking buffer) overnight, at 4 °C, and then washed four times in Tris-buffered saline with 0.1% of tween-20 (TBST) for 10 min. After washing steps, membranes were incubated with horseradish peroxidase-conjugated secondary antibody (diluted 1:3000 in 1 mL of blocking buffer) for an hour, at 4 °C. Following the wash steps as described above, the immunoreactive protein levels were detected by using ECL (Bio-Rad, Hercules, CA, USA), accompanied by a LAS-4000mini (Fuji Film, Tokyo, Japan).

### 2.10. Statistical Analysis

The statistic software GraphPad Prism 5 (GraphPad Software, Inc., San Diego, CA, USA) was used to calculate the differential significance, using Bonferroni’s Multiple Comparison Test. The significance difference (*p*-value) was acceptable as <0.05.

## 3. Results

### 3.1. Bio-Distributions of Bovine Serum Albumin–Coated Gold Nanoparticles (GNPs–BSA) in Gastric Cancer Xenografts

In order to track the tumor targeting of bovine serum albumin–coated gold nanoparticles (GNPs–BSA) in xenografts, GNPs–BSA nanoparticles were first pre-labeled with florescent dye Cy7.5 prior to intravenous tail vein injection into animals, followed by monitoring and tracing frutescence, using IVIS in vivo imaging system. The results revealed that fluorescence intensity appeared on tumors located on the animal’s right thigh and other areas (Figure 1A), suggesting the enhanced permeability and retention effect (EPR effect) induced by GNPs–BSA nanoparticles targeting tumors, but not in a very highly specific mode. GNPs–BSA nanoparticles were also retained in several organs, such as the liver, kidney, spleen, heart, lung, stomach, colon, and muscle from sacrificed animals (Figure 1B). Higher fluorescent intensity was detected and observed in kidney and liver compared to the other organs and tumors. Detoxification in the liver and urinary excretion in the kidney may contribute to the increased retentions of GNPs–BSA in these two major organs.

### 3.2. Bovine Serum Albumin (BSA) Severed as Coupling Agent of Tirapazamine (TPZ)

The hypoxia-induced toxic radical agent TPZ was covalently linked to GNPs for delivering cytotoxicity of hypoxic cancer and stromal cells for achieving a better therapeutic outcome. To process the conjugated TPZ and GNPs nanoparticles, BSA was applied as a coupling agent in this study. BSA molecular weight was ~66.9 kDa and acquired conjugated BSA–TPZ with a molecular weight of 74.2 kDa, as measured by a MALDI–TOF-MS analyzer, suggesting that TPZ was successfully conjugated with BSA (Figure 2A). For manufacturing conjugated GNPs–TPZ, BSA was applied as a coupling agent to link TPZ with GNPs. To bridge TPZ to GNPs, BSA was first conjugated with TPZ by adding glutaraldehyde into BSA- and TPZ-containing carbonate–bicarbonate solution (0.1 M) for the conjugate amino group (-NH_2_) of each BSA and TPZ. The Sephadex G-25 column was used to elute BSA–TPZ at fraction 2 by distilled water, and BSA appeared at 2~3 eluted fractions, while TPZ, measured at 450 nm, appeared at elution 5~6, indicating that BSA was successfully conjugated with TPZ. The conjugated efficacy was ~15%, which was calculated by dividing the amount of BSA–TPZ by total TPZ measured at 450 nm. (Figure 2B). It indicated that BSA was a potential TPZ binding agent for constructing the conjugated BSA-decorated GNPs–TPZ. The particle sizes were 69 ± 63 nm for pure GNPs, 104 ± 85 nm for GNPs–BSA conjugates, and 134 ± 75 nm for GNPs–TPZ conjugates determined by using dynamic light-scattering analysis (Figure 2C).

### 3.3. Increased Hydroxyl Radical (^•^OH) Released from TPZ under Hypoxia Condition

Tirapazamine (TPZ) is one of the hypoxic cytotoxins that selectively act in hypoxic tumor cells through bioreductive mechanisms. One-electron (1e^−^) reduction of TPZ by reductases leads to the formation of TPZ radical and may undergo homolytic cleavage to form the toxic species hydroxyl oxidizing radical (^•^OH) under hypoxia condition. Alternatively, TPZ can metabolically convert to oxidizing benzotriazinyl radical (BTZ). Both TPZ- and BTZ- radicals are DNA-damaging species and lead to the death of cells under a hypoxia environment. Hydroxyl oxidizing radicals give rise to the cytotoxic effects of cells resulting from DNA double-strand break (DSB), at least partially, associated with the topoisomerase-II dependent process [36,37]. In contrast to the hypoxia condition, the TPZ radicals are able to re-oxidize back to the original compound coupled with the production of a superoxide radical (O_2_^−^) if oxygen is present. To quantify and compare the amount of hydroxyl radical (^•^OH) released from TPZ either under hypoxic or normal (normoxia) conditions, 1 mg/mL of TPZ dissolved in PBS buffer was incubated at normoxia or anaerobic conditions (hypoxia), driven by an AnaeroPack system for producing ^•^OH. (Figure 3A) The ^•^OH release was detected by using an LTQ Orbitrap mass spectrometer by observing the structural exchange of TPZ from *m*/*z* 179 to *m*/*z* 162, which indicated that an ^•^OH escaped from TPZ. The relative intensity of *m*/*z* 162 to *m*/*z* 179 in hypoxia significantly increased compared to that in normoxia. The ^•^OH levels measured at OD 550 nm were also significantly elevated in hypoxia compared to that in normoxia (Figure 3B,C). Moreover, the value of OD 550 nm was increased in the GNPs–TPZ incubated in the hypoxia condition (Figure 3D). The results indicate that TPZ released a significantly higher amount of ^•^OH under anaerobic conditions.

### 3.4. Tumor Hypoxia Detection

Hypoxia-inducible factors (HIFs) are transcription factors regulated by oxygen-dependent dioxygenases and act as the primary oxygen sensors, contributing to a broad, adaptive response to hypoxia. HIF-1α is the hypoxia-response regulator of cells and is involved in the expressions of different genes for adapting to insufficient oxygen availability [38]. In response to growth factors, HIF-1α can be upregulated with increased protein expression levels through the activation of the PI3k/Akt-mTOR signaling pathway [39,40]. In addition, carbonic anhydrase 9 (CA9) is also one of the hypoxia-markers and transcriptionally regulated by HIF-1α. Therefore, we would like to evaluate the expressions of HIF-1α and CA9 on tumors or other organs of MKN45 xenograft mice for determining the hypoxic regions in animals. We imaged and quantified the expressions of CA9 in tumors and organs derived from animals by using Hypoxisense 680 Fluorescent Imaging Agent combined with an IVIS detection system [41]. In addition, the expression of HIF-1α expression, using electrophoresis combined with Western blotting, was assessed on tumor biopsy samples and organs, including lungs and kidneys. It revealed that the HIF-1α protein level in tumors was statistically significantly higher than in organs, including livers and kidneys (Figure 4A). We imaged and quantified the expression of CA9 in tumors at week 1 (1W) and week 4 (4W), demonstrating a significant increase in CA9 expression in 4W tumors (Figure 4B). Besides tumors, CA9 expressions in the stomach and colon were significantly higher than in other organs, such as the liver, kidney, spleen, heart, lung, and muscle (Figure 4C). These results indicated that hypoxic markers, such as HIF-1α and CA9, were highly expressed on the tumors and their expressions were increased with tumor growth, suggesting the presence of hypoxia in tumor tissues and hypoxic degree increased with tumor progression.

### 3.5. Effect of Tirapazamine (TPZ) on Cell Viability of Gastric Cancer Cells

To compare the cell viability of MKN45 gastric cancer cells exposed to TPZ either under normoxia or hypoxia condition, cells first were pre-cultured in a medium containing 1, 10, and 100 μg/mL of TPZ, followed by cell viability assessment, using WST-1 assay. The results indicated that cell viability was significantly reduced in cells treated with TPZ at ≥10 μg/mL under normoxia condition and ≥1 μg/mL under hypoxia condition, suggesting that hypoxia induced the enhanced cytotoxicity of TPZ on MKN45 gastric cancer cells (Figure 5A). An increased expression of catalase and glutathione-S-transferase Pi 1 (GSTP1) in cells exposed to TPZ at 1 and 5 μg/mL under hypoxia conditions was detected by using Western blots (Figure 5B). The quantification of increased catalase and GSTP1 from MKN45 cells treated with 1 or 5 μg/mL under hypoxia conditions is depicted in Figure 5C,D, respectively. GSTP1 belongs to the family of glutathione S-transferases (GSTs) with the enzymatic activity on detoxification [42]. Catalase is an antioxidant enzyme in all aerobic organisms to catalyze hydrogen peroxide into oxygen and water.

### 3.6. Therapeutic Assessment of BSA-Decorated GNPs–TPZ (GNPs–TPZ) on MKN45 Xenograft

Previous studies indicated that TPZ increased higher expressions of cell-stress factors under hypoxia tumor niche, contributing to cytotoxicity against MKN45 cells. Therefore, BSA-decorated GNPs provided a great shuttle device to carry and concentrate TPZ into the hypoxic tumor through EPR effect for cancer cell eliminations, but caused no harm to normal cells residing in the normoxia. BSA-decorated GNPs–TPZ may achieve the best therapeutic outcome with minimized side effects. To evaluate the therapeutic effects of BSA-decorated GNPs–TPZ on MKN45 gastric cancer cells, we first performed GNPs–TPZ-mediated inhibitory effects on cells incubated either under normoxia or hypoxia conditions. The results revealed that TPZ significantly reduced the cell viability of MKN45 gastric cancer cells either under normoxia or hypoxia conditions (Figure 6A). Notably, TPZ imposed a much higher inhibitory effect on the cell viability of cells cultured in the hypoxia than ones incubated in the normoxia condition. The in vivo therapeutic assessment of GNPs–TPZ on the MKN45 xenograft animals was performed, and the results revealed that GNPs–TPZ exhibited the best therapeutic outcome with the smallest tumor volume in animals (Figure 6B). To unbraid the mechanism in reducing the tumor volume mediated by GNPs–TPZ in MKN45 xenograft animals, we first examined biopsy samples, including tumors and liver, which were surgically removed from sacrificed animals by using TUNEL assay (Terminal deoxynucleotidyl transferase dUTP nick end labeling). Compared to a tumor, considerable low fluorescent intensity was detected in the liver. In contrast, fluorescence was detected in the tumor tissues from mice treated with GNPs, TPZ, and GNPs–TPZ (Figure 6C). Moreover, it showed the highest intensity in GNPs–TPZ-treated tumor samples, indicating that GNPs–TPZ mediated DNA fragmentation, a hallmark of apoptosis, in tumors.

### 3.7. Assessment of Adverse Effects of GNPs–TPZ

Many concerns have been raised regarding the applications of GNPs in the biomedical field, such as cytotoxicity (side effects) and systemic bio-distribution. Some studies suggested that accumulated nanoparticles impaired the physiological functions of several major organs. However, other studies indicated that gold nanoparticles at a low amount were considerably safe and suitable for the enhancement of radiotherapy, photothermal therapy, and related medical diagnostic procedures [43,44]. We measured and compared the biochemical and inflammatory activity of the following proteins in the serum of mice exposed to TPZ, GNPs, TPZ, and GNPs–TPZ for evaluating the changes in functions. The proteins, such as aspartate aminotransferase (ALT), alanine transaminase (AST), creatinine, and high-sensitivity C-reactive protein levels, were measured and compared in mice, and the results demonstrated that there was no significant difference in the statistical assessment between GNPs-TPZ and normal group. The AST and ALT levels in mice exposed to GNPs were relatively lower than other groups and also fell out of the minimal normal range of mice. In addition, there were no significant differences in the level of creatinine and HS-CRP among groups, and all were under the normal range of mice (Table 1).

## 4. Discussion

With the giant progress in developing nanotechnology, the knowledge and tools of nanomedicines have been applied in the fields of diagnostics and pharmaceuticals for the prevention and treatment of diseases. Nanoparticles benefit the loaded poorly soluble hydrophobic drugs with improved pharmacokinetics by solubilizing them in the hydrophobic compartments. Furthermore, nanoparticles assist chemotherapeutic drug delivery by inherent passive targeting phenomena and adopted active targeting strategies [45]. In addition to targeted therapy drugs supplemented with tumor-specific surface markers, hypoxic cytotoxins, such as Tirapazamine, also exhibit the preference in tumor targeting and selectivity in cancer cell killings. Increased accumulation and penetration of hypoxic cytotoxin cancer drugs to the hypoxic center of the tumor is often challenging due to the large inter-capillary distances and variable blood flow of solid tumors [46,47,48]. To solve these difficulties in delaying or inhibiting drug targeting to tumor cells and provide more therapeutic efficiency for hypoxic cytotoxin, gold nanoparticles are great candidates to assist TPZ in passing these barriers and reach the hypoxic tumor cells through the enhanced permeability and retention effect (EPR) [49].

Recently, bovine serum albumin (BSA) emerged as one of the common coupling agents for decorating or bridging nanoparticles to functional drugs for tissue or cell targeting. Besides being used as a coupling agent, bovine albumin can facilitate the endothelial transcytosis of unbound and albumin-bound plasma constituents to extravascular space, leading to enhanced penetrations and accumulation of drugs into tissues [50]. Considering the merit in the safety of gold nanoparticles, GNPs have become a favorable platform for many applications in medicine, because of the established nontoxicity of the gold core [51]. To combine the advantages of GNPs and BSA, we generated the conjugated GNPs–TPZ nanoparticles, using coupling agent BSA to enrich TPZ in penetrations and aggregations of hypoxic tumors. Followed by the preparations of GNPs–BSA nanoparticles, GNPs–BSA was labeled with fluorescence Cy7.5 for tracing its bio-distributions in xenograft. The results displayed the fluorescence-labeled GNPs–BSA with adequate systemic bio-distributions in animals with strong fluorescent intensity in the tumor region (Figure 1B), suggesting that the conjugated GNPs–BSA nanoparticles accumulated in tumors and were a suitable candidate as a drug delivery vehicle for cancer therapy of TPZ hypoxic cytotoxin. In addition to tumors, several organs, including the liver and kidney, also exhibited high fluorescent intensity, suggesting that GNPs–BSA displayed a higher affinity for tumors, liver, and kidney (Figure 1B). This finding is consistent to the previous study, indicating that increased accumulations of GNPs were detected in liver and kidney after intravenous administrations of gold nanoparticles [52].

Due to the availability and the existence of multiple and broad functional chemical groups to interact with, serum proteins have become popular coupling agents for covalently interacting with drugs or targeted markers via disulfide bonds or others [31,53,54]. In this current study, we were the first to generate BSA-coated TPZ. To confirm and certify these acquired BSA–TPZ nanoparticles, several assessment tools were used, including a MALDI–TOF-MS analyzer, Sephadex G-25 column, and Bio-Rad Protein Assay. Bovine serum albumin is the most abundant protein in the plasma, with a molecular mass of 66 kDa [55]. After the synthetic and coupling process, the BSA–TPZ exhibited the molecular weight of 74.2kDa, which is significantly higher than BSA alone, with a molecular weight of ~66.9 kDa (Figure 2A). These significant changes in molecular weight, as measured by the MALDI–TOF-MS analyzer, suggested that the binding of bovine serum albumin to the anticancer agent, TPZ, was successfully achieved. Followed by molecular weight monitoring and comparison, the BSA–TPZ and others were eluted from a mixture, using the Sephadex^®^ G-25 gel filtration column, and determined by using a Bio-Rad Protein Assay. The results indicated BSA–TPZ was a collection at elusion 2, BSA was collected between elution 2 and 3, and TPZ was collected between elution 5 and 6 (Figure 2B). Moreover, the completed form of the conjugated BSA-coated GNPs–TPZ displayed an average diameter of ~134 ± 75 nm, which was measured by using dynamic light-scattering analysis; the diameter was larger than that of GNPs and GNPs–BSA, at 69 ± 63 nm and 104 ± 85 nm, respectively (Figure 2C).

To confirm our hypoxic cytotoxin TPZ with the ability to release or convert the TPZ-related hydroxyl radical (^•^OH), the ratio of 162/179 *m*/*z* was measured and compared between two forms of TPZ, either under normoxia or hypoxia, using LTQ mass spectrometry, for representing the hydroxyl radical (^•^OH) levels (Figure 3A), indicating there were more hydroxyl radical (^•^OH) detected under hypoxia condition quantified by using LTQ orbitrap mass spectrometer. After an additional measurement using an ROS assay kit to confirm the hydroxyl radical (^•^OH) release, the increased intensity of absorbance at 550 nm under normoxia condition was observed, suggesting a significantly increased amount of ^•^OH release under the anaerobic condition. Meanwhile, GNPs–TPZ exhibited a higher ^•^OH release in hypoxia condition.

We also would like to unbraid the hypoxia region in the MKN45-induced xenografts for further evaluations of TPZ in in vivo hypoxia targeting. Two hypoxia-stress markers, namely HIF-1α and carbonic anhydrase 9 (CA9), were used. The results revealed the increased expression in tumors but extremely low expression of HIF-1α in the lungs and kidneys. Moreover, the real-time imaging of the expression of CA9, using a HypoxiSense 680 reporting fluorescent agent, was largely detected on tumors, and the fluorescent intensity became significantly stronger with the tumor growth based on the comparisons of tumor volume between week 1 and week 4 after tumor implantations. To detail the organ distributions of CA9, several organs were resected and quantitatively compared in the fluorescence intensity. The fluorescence intensity indicated that CA9 is higher expressed on tumors and also on two other organs, namely the stomach and colon. A higher amount of CA9 mRNA expression in gastric tissue and colon in mice has been reported, and this is consistent with our result illustrated in Figure 4B [56].

Followed by the detection of the hypoxic region in animals, the cytotoxicity of TPZ in cells was characterized. It indicated that TPZ amplified cytotoxicity coupled with decreased cell viability of MKN45 cells in hypoxia; moreover, the degrees in reductions of cell survivals were positively correlated to the dosage of TPZ applied in the medium in a dose-dependent manner. This confirmed that TPZ is a hypoxic cytotoxin, as described in the previous literature. We further investigated whether TPZ induced oxidative stress in MKN45 cells by dissecting the expression of the lead oxidative-stress proteins, such as catalase and GSTP1, which play an important role in detoxification. The results demonstrated that TPZ induced the oxidative stress of MKN45 cells by elevating the expression of catalase and GSTP1 for processing the subsequent detoxification.

The main goal of this study was to determine the therapeutic effect of conjugated GNPs–TPZ nanoparticles. To achieve this therapeutic assessment, we first tested the cytotoxic effect of gastric cancer cells, and then in the MKN45-induced xenograft animal model with the administration of GNPs–TPZ nanoparticles, the results indicated that GNPs–TPZ remained to keep the property of TPZ as a hypoxic cytotoxin with inhibitory effects on cell viability. Indeed, GNPs–TPZ induced significant reductions in cell viability of MKN45 cells under hypoxia compared to cells cultured in normoxia. The results from in vivo therapeutic assessment in xenografts were consistent with our findings from in vitro studies on cell viability. It revealed that the smallest nodule in average tumor volume was observed in animals treated with GNPs–TPZ among other experimental groups, including mice with administration of PBS, gold nanoparticles (GNPs), or TPZ, suggesting that the GNPs–TPZ provides the best therapeutic outcome. To unbraid the TPZ-mediated molecular mechanism in inhibiting or reducing tumor growth, tumors and liver were resected and fixed for processing TUNEL assay. It revealed the significantly higher fluorescence intensity on tumor tissue with exposure to GNPs–TPZ nanoparticles, but a relatively dim fluorescence intensity appeared in the liver, suggesting that GNPs–TPZ targeted and eradicated hypoxic cancer cells through apoptosis cell-death program.

We also examined the potential GNPs–TPZ-mediated side effects in xenograft by collecting and assessing the biochemical and inflammatory activity of AST, ALT, creatinine, and HS-CRP from blood samples. It indicated that the value of AST, ALT, and creatinine derived from blood samples stayed in the normal range and showed no statistical difference compared to control mice, indicating its safety. Furthermore, the HS-CRP levels were considerably close and showed no statistical differences, suggesting that no inflammation or cardiovascular disorders or malfunctions occurred in mice that received treatments either with TPZ, GNPs, or GNPs–TPZ (Table 1). Taken together, GNPs–TPZ did not alter physiological functions in the liver, kidney, and heart of animals.

## 5. Conclusions

We found that GNPs are a great carrier for TPZ in improving tumor accumulations and that BSA was a suitable binding agent to facilitate the formation of the conjugated GNPs–TPZ particles. The conjugated GNPs–TPZ particles with the size of 134 ± 75 nm exhibited great power in tumor accumulation via an EPR effect. The GNPs–TPZ–mediated cytotoxic molecular mechanism was turned on under the hypoxic condition, such as hypoxic tumors, by releasing hydroxyl oxidizing radicals to kill cancer cells, leading to the reduced tumor volume of MKN45-derived tumor xenografts. Taken together, the conjugated GNPs–TPZ particles exhibited enhanced therapeutic efficacy in MKN45-derived tumor xenografts, thus suggesting that these particles are effective and safe nanomedicines with great specificity to target hypoxia tumors. The conjugated GNPs–TPZ nanomedicines may be a potent candidate for cancer treatments, and further preclinical studies should be taken into consideration in the future.

## Figures and Tables

**Figure 1 pharmaceutics-14-00847-f001:**
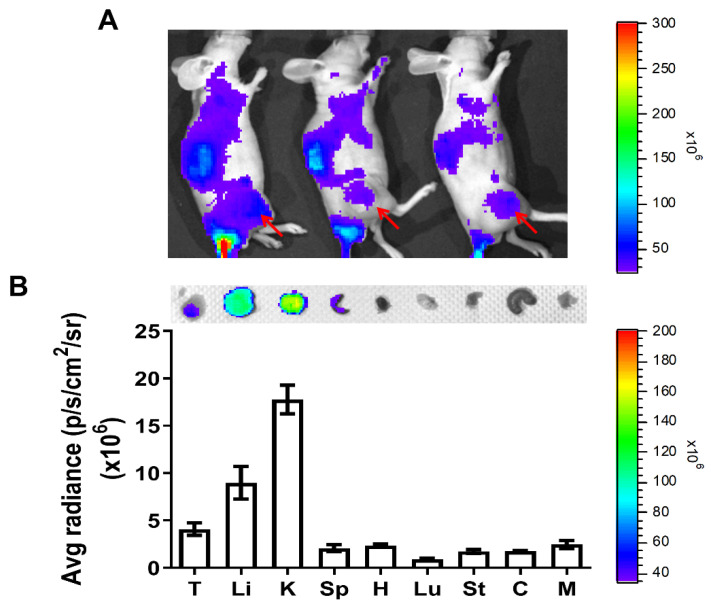
Bio-distributions of bovine serum albumin–coated gold nanoparticles (GNPs–BSA) in gastric cancer xenografts. To measure the tissue distributions of GNPs–BSA nanoparticles, the Cy7.5 pre-labeled GNP–BSA were intravenously injected into animals for 48 h and then analyzed by using in vivo imaging system (IVIS). The images revealed that fluorescent GNPs–BSA nanoparticles were not only distributed in tumors located on the right thigh, but also in other regions, as shown in (**A**). The fluorescent intensity in the major organs, including liver (Li), kidney (K), spleen (Sp), heart (H), lung (Lu), stomach (St), colon (C), and muscle (M), was also measured and compared. Quantitative data were determined by the valve of fluorescent intensity/tumor volume (*n* = 3). Average radiance values of GNPs–BSA in the organs and tumors shown in (**B**). p/s/cm^2^/sr: photons per second per square centimeter per steradian.

**Figure 2 pharmaceutics-14-00847-f002:**
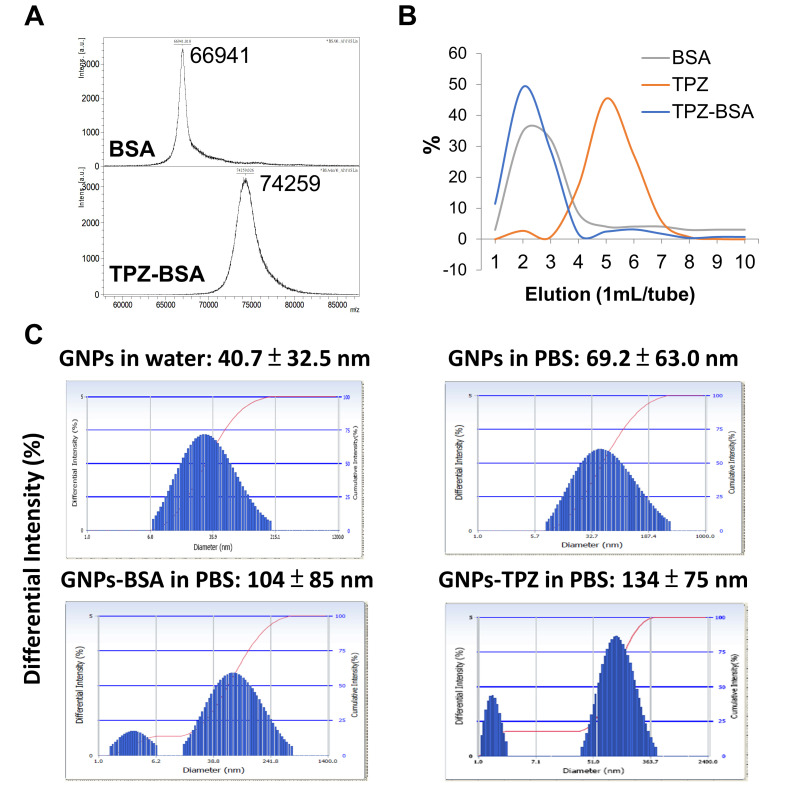
Bovine serum albumin (BSA) severed as coupling agent of TPZ. The molecular weight of BSA was measured as ~66.9 kDa, and the conjugated BSA–TPZ was 74.2 kDa measured by using a MALDI–TOF-MS analyzer (**A**). The conjugated BSA–TPZ was collected in elution 2 and measured at 450 nm, using a Sephadex G-25 column, and confirmed by Bio-Rad Protein Assay (**B**). The distributions in size were measured by using dynamic light-scattering analysis. Particle sizes were 69 ± 63 nm for pure GNPs, 104 ± 85 nm for GNPs–BSA conjugates, and 134 ± 75 nm for GNPs–TPZ conjugates (**C**).

**Figure 3 pharmaceutics-14-00847-f003:**
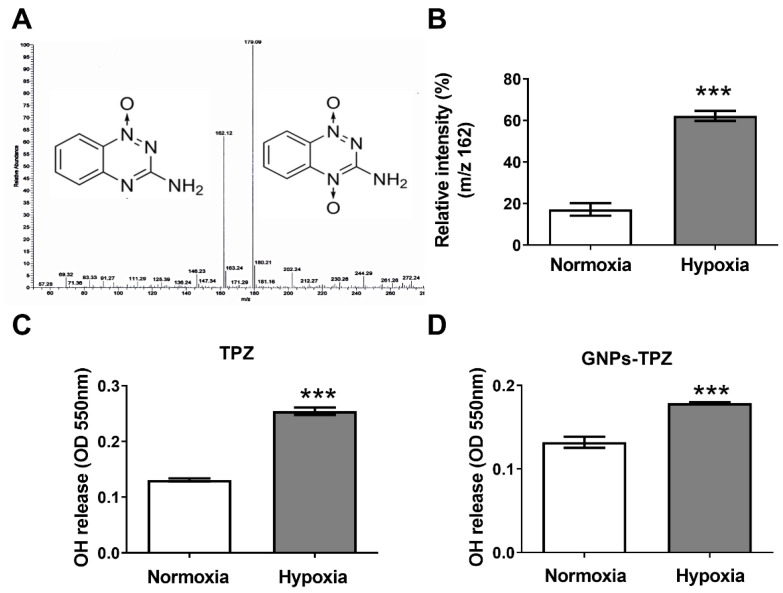
Increased hydroxyl radical (^•^OH) released from TPZ under hypoxia condition. Relative intensity measurement of TPZ hydroxyl radical (^•^OH) release in hypoxia condition. Ratio 162/179 *m*/*z* was measured by using LTQ mass spectrometry for representing the OH levels (**A**) The comparison in the relative intensity of hydroxyl radical (^•^OH) released from TPZ was measured in hypoxia and normoxia conditions quantified by using LTQ Orbitrap mass spectrometer (**B**). An increased absorbance at OD 550 nm correlated to hydroxyl radical (^•^OH) released from TPZ was measured and compared in hypoxia and normoxia conditions (**C**). In addition, the value of OD 550 nm was measured in GNPs–TPZ incubated in hypoxia compared to that in normoxia (**D**). *** *p* < 0.001.

**Figure 4 pharmaceutics-14-00847-f004:**
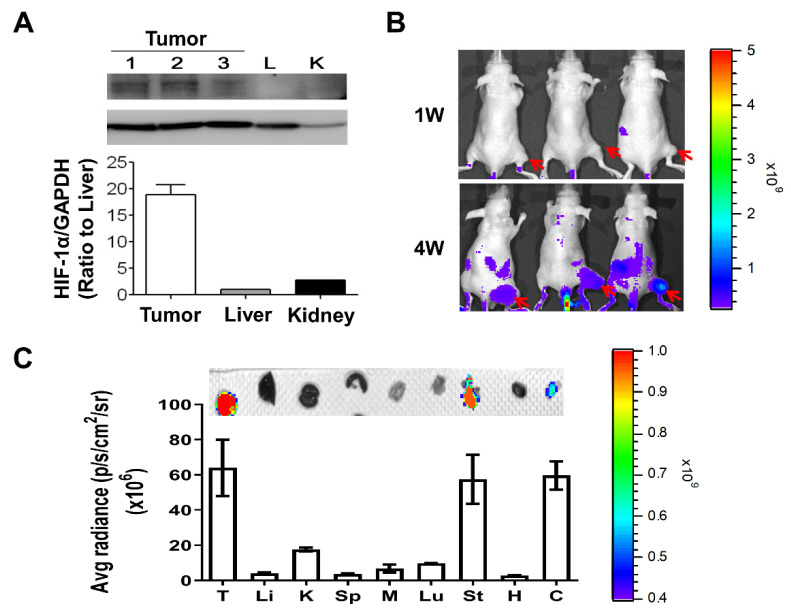
Tumor hypoxia detection. Two hypoxia markers, namely HIF-1α and carbonic anhydrase 9 (CA9), were used for the detection of tumor hypoxia in the MKN45-induced xenograft animal model. Increased HIF-1α expression in tumors of MKN45-induced xenografts compared to organs, such as lungs and kidneys (**A**). The direct in vivo imaging and detection of CA9 expression in MKN45-induced xenograft after 24-h injection of HypoxiSense 680 agent was depicted in (**B**) (*n* = 3). The average fluorescent density of CA9 was detected and compared in tumors and major organs, including in liver (Li), kidney (K), spleen (Sp), muscle (M), heart (H), lung (Lu), stomach (St), and colon (C), and depicted in (**C**). p/s/cm^2^/sr: photons per second per square centimeter per steradian.

**Figure 5 pharmaceutics-14-00847-f005:**
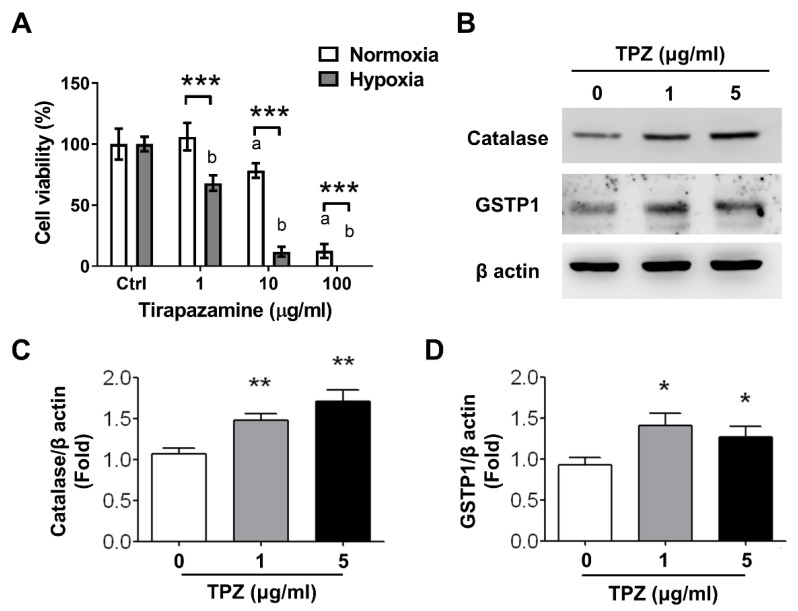
Effect of Tirapazamine (TPZ) on cell viability of gastric cancer MKN45 cells. The viability of MKN45 cells exposed to PBS only (indicated in Ctr) or TPZ at 1, 10, or 100 µg/mL for 48 h was measured by using WST-1 assay (**A**). The lowercase letter “a” indicates statistical significance compared to control under normoxia, and the “b” indicates statistical significance compared to control under hypoxia; *** *p* < 0.001 represents statistical significance between the labeled groups. TPZ aggravated the reduction in cell viability of MKN45 gastric cancer cells incubated under hypoxia (**B**). TPZ increased the expression of oxidative-stress-leaded enzymes, such as catalase and GSTP1 (**C**,**D**); ** *p* < 0.01 and * *p* < 0.05 represent statistical significance between the labeled groups.

**Figure 6 pharmaceutics-14-00847-f006:**
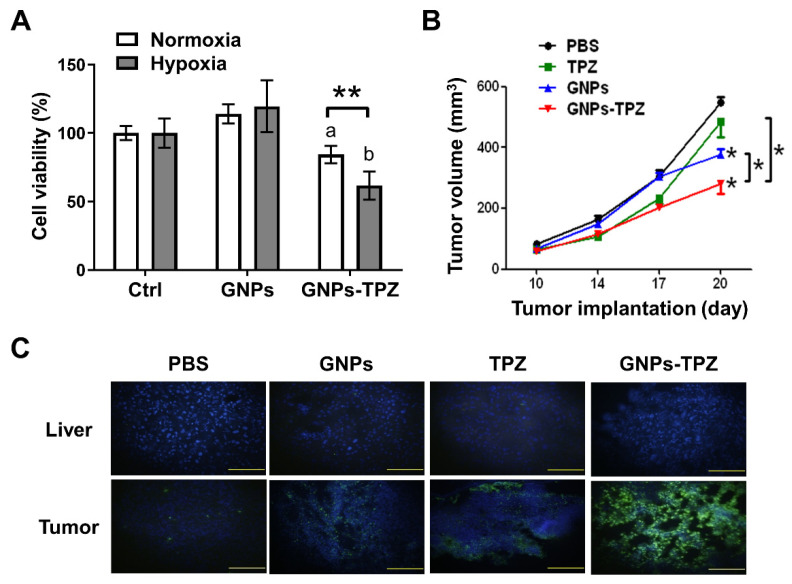
GNPs–TPZ treatment contributed to the reduction of MKN45 cell viability and tumor volume of MKN45 tumor-bearing xenografts. A significant reduction in cell viability of MKN45 cells cultured in medium containing GNPs–TPZ under hypoxia for 48 h compared to in normoxia (**A**). Cell viability was assessed by using WST-1 assay; ** *p* < 0.01 represented the statistical significance of cells receiving GNPs–TPZ treatment in hypoxia vs. normoxia. The lowercase letter “a” indicates statistical significance compared to control under normoxia, and the “b” indicates statistical significance compared to control under hypoxia. Changes in tumor volume in MKN45 xenografts post-intravenous injection of PBS, TPZ, GNPs, and GNPs–TPZ at the dose of 0.5 mg/kg (*n* = 3 for each group) for 20 days were monitored and compared. As implanted tumors reached 50 mm^3^, drug injections were carried out at days 10, 14, and 17 (**B**); * *p* < 0.05 represented statistical significance among the labeled groups. Histological apoptosis detections in tumor and liver tissues resected from xenografts post-treated with PBS, Free GNPs, TPZs, and GNPs–TPZ by TUNEL staining. Green, apoptosis DNA; blue, cell nuclei. Scale bar: 100 μm (**C**).

**Table 1 pharmaceutics-14-00847-t001:** Biochemical and inflammatory activity in liver and kidney functions.

Parameters (Normal Range)	AST (U/L) (59~247)	ALT (U/L) (28~132)	Creatinine (mg/dL) (0.2~0.8)	HS-CRP (mg/dL)
Normal	66.67 ± 4.51	12.67 ± 1.16	0.38 ± 0.09	0.07 ± 0.03
TPZ	60.00 ± 14.13	10.67 ± 1.53	0.32 ± 0.02	0.08 ± 0.01
GNPs	32.33 ± 7.57 *	8.00 ± 2.65	0.34 ± 0.01	0.10 ± 0.01
GNPs–TPZ	58.67 ± 19.30	14.33 ± 8.39	0.38 ± 0.06	0.09 ± 0.00

Aspartate aminotransferase, alanine transaminase, creatinine, and high-sensitivity C-reactive protein levels in mice after administration of TPZ, GNPs, and GNPs–TPZ; * *p* < 0.05 compared to normal, TPZ, and GNP-TPZ in AST level.

## Data Availability

Not applicable.

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
