# Peer review of "Improving Tirapazamine (TPZ) to Target and Eradicate Hypoxia Tumors by Gold Nanoparticle Carriers"

_pharmaceutics, 2022, doi:10.3390/pharmaceutics14040847_

Round 1

Reviewer 1 Report

The authors presented their research very well. I have few questions below.

  • How did the authors optimize the concentration of glutaraldehyde
  • Please explain stability studies of prepared GNPs
  • Plese expand conclusion part.

Author Response

First of all, thank you for your comments and suggestions that allowed us to greatly improve the quality of this manuscript. We revised this manuscript with updated data in figure 2C and figure 3D.  The manuscript was edited by an English speaker native. We also added one reference  ( citation 35) for the response to a question regarding optimizing the concentration of glutaraldehyde in material and method.

Please see the attached file with the detailed response to your concerns and questions.

Reviewer 2 Report

The article titled “Improving tirapazamine (TPZ) to target and eradicate hypoxia tumors by gold nanoparticle carriers” may be a useful contribution to the journal; however, I recommend a minor revision should be taken into consideration before publishing the work.

  1. Please check the abbreviated term throughout the article and maintain a unique representation.

For eg., in Figure 2 caption the author measured the size of GNPs-BSA-TPZ and abbreviated the term; but there was no such term was given in Table 1.

It must be corrected to give better clarity to the readers.

  1. Please provide the error bars (±) to the Figure 1B; Figure 3B,C; Figure 4A,C; Figure 5A,C,D; and Figure 6A and in the current version it indicates only the + values of errors.
  2. Figure 6C - Please provide scale bar value
  3. Figure 2C captions – The author mentioned that the size of conjugated GNPs-BSA-TPZ ranged from approximately 6.0 to 215.1 nm with the mean diameter of ~30 nm. I believe the experiment was performed in triplicate, please provide the average value of size including the ± variations.
  4. Please elaborate on the conclusions (Section 5) by briefly, adding the concluding remarks of the studies performed using GNPs-BSA-TPZ.

Author Response

First of all, thank you for your comments and suggestions that allowed us to greatly improve the quality of this manuscript. We revised this manuscript with updated data in figure 2C and figure 3D.  The manuscript was edited by an English speaker native. 

Please see the attached file with the detailed response to your concerns and questions.

Reviewer 3 Report

This work used gold nanoparticles-tirapazamine (GNPs-TPZ) conjugate followed by evaluating its therapeutic effects on the MKN45 induced xenografts. The work is still premature to be accepted:

1) Section 3.2, the authors claimed that “BSA molecular weight was ~66.4 kDa and the acquired BSA-TPZ conjugates with molecular weight was 74.2 kDa”. Then how many TPZ molecules are conjugated to each BSA molecule? Moreover, this result is shown as 72.4 kDa in several places.

2) Section 2.3, how can Sephadex G-25 column be used to purify macromolecules, since this column is a typical desalting column?

3) The TEM images for GNPs, GNPs-BSA and GNPs-TPZ conjugates should be provided.

4) Section 3.4 and 3.5, Fig. 4 and 5 show some known results before and after treatment with TPZ. What about the case for GNPs-TPZ conjugate? Note that this conjugate should be focused in this research, which was not emphasized in the manuscript.

Author Response

(The authors gave the same response as above.)

Round 2

Reviewer 3 Report

The authors have improved the manuscript. However, I think more details are required. For example, 1) does the control group (ctrl) in Figure 5A mean a group using TPZ of 0 μg/mL?  2) the original photos for different mice treatment groups should be added to verify the results shown in Figure 6B.

Author Response

Thanks for your comments and suggestions. We revised this manuscript to improve the quality of this study. In addition to the response in the attached file, we updated the information in the figure legend of Figure 5A. Please see the detail in the revised manuscript and the attached file. Again, thanks for your insightful considerations.
